# Impact of the Urban Exodus Triggered by the COVID-19 Pandemic on the Shrinking Cities of the Osaka Metropolitan Area

**Haruka Kato *** and **Atsushi Takizawa**

Department of Housing and Environmental Design, Graduate School of Human Life Science,
Osaka City University, Osaka 5588585, Japan; takizawa@osaka-cu.ac.jp
* Correspondence: haruka-kato@osaka-cu.ac.jp; Tel.: +81-6-6605-2823

**Abstract:** This study aims to clarify the impact of the urban exodus triggered by the COVID-19 pandemic on shrinking cities in the Osaka metropolitan area, where a declining population is caused by population aging. Analyzing the Osaka metropolitan area enables us to clarify how cities are shrinking due to the urban exodus. This study analyzed the monthly population data of three types of municipalities: ordinance-designed/regional hub cities, ordinary cities, and towns/villages. In conclusion, the study clarified that population change due to the urban exodus occurred in the ordinance-designed/regional hub and ordinary cities from summer to autumn 2020. The most significant population increases occurred in the municipalities in the Osaka metropolitan fringe area, which are located more than 30 km away from the center of the Osaka metropolitan area. The conclusion is important because the population increased not only in the ordinance-designed cities but also in the ordinance-designed/regional hub cities, unlike the rest of the metropolitan area. The result is the new insights unique to the Osaka metropolitan area that this study clarified. The urban exodus contributes to the need for the local governments of shrinking cities to maintain the urban services necessary for people's daily lives.

**Keywords:** population; shrinking cities; urban exodus; COVID-19 pandemic; Osaka metropolitan area

## 1. Introduction

Shrinking cities, which are cities with declining populations, are a significant issue in urban planning. Well-known examples include the urban planning of the Rust Belt in the United States, where the shrinking of cities increased due to economic decline, and Germany, where the shrinking of cities increased due to political changes caused by the integration of East and West Germany [1]. In Japan, population decline due to aging has become a significant issue for urban planning [2]. In 2020, the Japanese population aged over 65 years old was 36.19 million, constituting 28.8 percent of the total population [3]. As in China, research on shrinking cities has progressed in East Asia [4]. All over the world, many factors of shrinking cities have been studied, such as economic decline, political change, and the aging population.

This study's research question is as follows: Is the urban exodus that is triggered by the COVID-19 pandemic a factor in the shrinking of cities in metropolitan areas? The COVID-19 pandemic, which started in Wuhan, has spread worldwide. In order to prevent the spread of the COVID-19 infection, lockdown is one of the most effective measures [5]. Many cities worldwide have implemented lockdowns, which have restricted the activities of people living in city centers. Restrictions on daily activities include orders to stay at home, requests to work from home, closure of cultural and entertainment facilities, and restrictions on eating and drinking inside restaurants. The lockdowns enforced more substantial restrictions on the daily activities of people living in areas closer to the city center [6]. Most urban populations were forced to live inconveniently with social distancing policies in place, if not a full lockdown. McGrail et al. [7] found that social distancing

policies significantly reduced the COVID-19 spread rate over two weeks in 134 countries worldwide. However, most urban populations worldwide faced travel restrictions, even within their own country [8].

Therefore, the phenomenon of the urban exodus arose in cities worldwide [9]. The urban exodus is mass migration out of urban areas to avoid difficulties [10]. During the COVID-19 pandemic, urban exodus might have caused a new type of shrinking city. As the impact of the COVID-19 pandemic continues and becomes prolonged, it is essential to analyze the urban exodus to develop urban policies that coexist with the COVID-19 pandemic.

This study aimed to clarify the impact of the urban exodus triggered by the COVID-19 pandemic on shrinking cities. For this purpose, the study analyzed the Osaka metropolitan area in Japan as a case study. The Japanese state of emergency was called the soft lockdown [11]. That is because the Japanese government did not restrict the activities of individuals [12]. The case of Japan, where restrictions were looser than in other countries, contributes, for policymakers of other countries, to consider relaxing some of their restrictions. In Japan, the Osaka metropolitan area has a declining population caused by population aging [13]. Therefore, analyzing the Osaka metropolitan area enables policymakers to determine the impacts on shrinking cities of the urban exodus due to the COVID-19 pandemic.

Regarding the urban exodus triggered by the COVID-19 pandemic, this study reviewed not only articles but also commentaries, reports and preprints. This is because urban exodus has just begun to be studied around the world. The driving factor of the urban exodus is the change of people's residential location choice preferences. Before the COVID-19 pandemic, the residential location choice preferences were workplace location, including commuting transportation [14,15]. However, during the COVID-19 pandemic, Nathan et al. [16] suggested that workplace location might weakly influence the choice of residence. That means that people tend to choose their residence based on factors other than the location of their workplace. For example, in Japan, Mitsubishi UFJ Research and Consulting Co., Ltd. [17] reported that more people placed importance on medical and welfare facilities and the surrounding natural environment in choosing their residence by a web questionnaire survey in January 2021. That change might be related to the lockdown and the collapse of medical systems in some metropolitan areas. Therefore, in China, Jia et al. [18] clarified that massive population movements occurred by 24 January 2020, when the lockdown began, with a correlation between the population flow and the number of infected people [18]. Massive urban exodus also occurred outside of China. In the United States, a correlation was found between population mobility and COVID-19 transmission [19]. In New York City and Boston, approximately 20% of the population moved away in April 2020 compared to the previous year [10]. Many people who moved out of the New York City area moved to neighboring cities in the metropolitan area [10]. In India, the rural population increased by 7%, while the urban population decreased by approximately 4 to 11% [20]. The urban exodus to rural areas is sometimes criticized for spreading the infection to rural areas. However, Weisbuch [21] clarified that the number of infected people does not expand in rural areas due to factors, such as having more opportunities to use cars and fewer stores. In a different trend, in Italy, during the first wave of the pandemic, the population decreased in the urban fringe area [22]. The population decline could be related to Italy's decentralized transportation network and urban structure [23]. These studies suggest that the areas where the population increases or decreases due to urban exodus vary from country to country. Therefore, the novelty of this study is to clarify the urban exodus in the Osaka metropolitan area. This study provides valuable insights into urban policies for shrinking cities.

This manuscript consists of five chapters based on the IMRaD: materials and methods in Section 2; results in Section 3; discussion in Section 4; and conclusion in Section 5.

## 2. Materials and Methods

### 2.1. Population Data

The data of this study are the monthly population statistics of municipalities. This study utilized the basic resident registration, a public register created by each municipality based on the Residential Basic Book Act [24]. Using the Residential Basic Book Act, each prefecture publishes the population on the first day of each month. This study analyzed the total populations of each municipality of Osaka [25], Kyoto [26], Hyogo [27], Nara [28], Shiga [29], and Wakayama [30] prefectures.

The Japanese government classified these municipalities into four categories according to their population size and role in the metropolitan area [31]. The categories are ordinance-designated cities, regional hub cities, ordinary cities, and towns/villages. The ordinance-designated cities are municipalities that are designated by the central government to be cities with a population over 500,000. The ordinance-designed cities have the authority to establish wards and use some authority similar to that of prefectures. The ordinance-designated cities include wards of Osaka, Sakai, Kyoto, and Kobe cities in the Osaka metropolitan area. The regional hub cities are municipalities designated by the central government to be cities with a population over 200,000. The regional hub cities have some authority similar to prefectures, but only in specific fields, such as health and urban planning. Ordinary cities are municipalities with populations over 50,000. Towns/villages are other municipalities where are located in rural areas. This study analyzed ordinance-designated and regional hub cities together as "ordinance-designated/regional hub cities" because the ordinance-designed and regional hub cities have similar roles in Japan. This study compared ordinance-designed/regional hub cities (N = 66), ordinary cities (N = 92), and towns/villages (N = 87) in Figure 1.

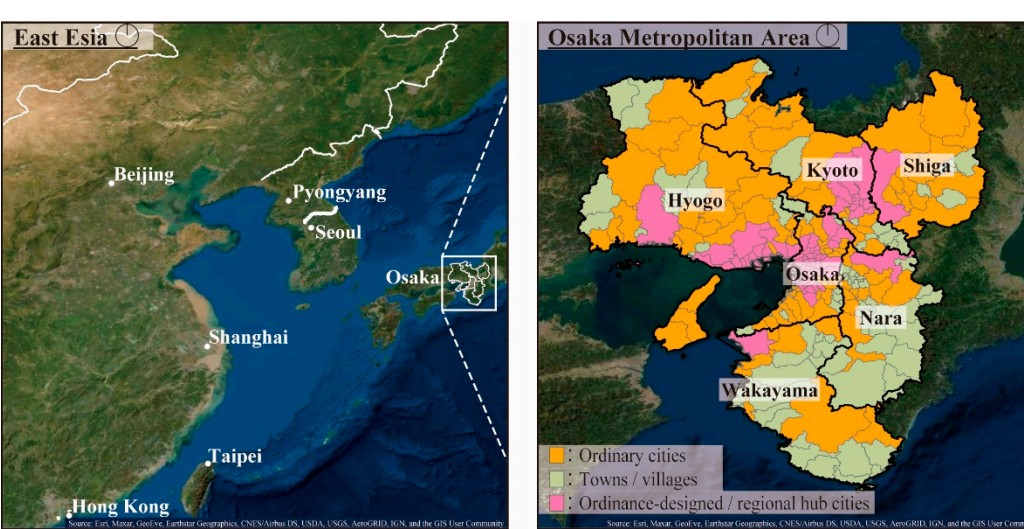

**Figure 1.** Map of Osaka metropolitan area.

### 2.2. Analysis Period

The analysis period for this study is from January 2017 to September 2021. On 14 January 2020, the World Health Organization (WHO) confirmed the onset of a coronavirus. Then, Wuhan entered lockdown from 23 January 2020. Then, the WHO named the new coronavirus COVID-19 on 11 February 2020, and declared a global pandemic on 11 March 2020. Therefore, this study analyzes population changes from January 2017 to December 2019 as the pre-pandemic and from January 2020 to September 2021 as the pandemic.

Figure 2 summarizes the number of people infected with SARS-CoV-2 in the Osaka metropolitan area. The data were obtained from Japan Broadcasting Corporation's daily data on the number of infected people in each prefecture [32]. Emergency declarations were

made four times in the Osaka metropolitan area. The first state of emergency was declared in all prefectures. After the end of the declaration, to recover the economy, the Japanese government began the "Go-To Travel" campaign for the hotel and restaurant industries [33]. However, the number of infected people gradually increased [34]. Therefore, the second and subsequent states of emergency were declared mainly in Osaka, Kyoto, and Hyogo prefectures in 2021. Vaccination started for healthcare workers in February 2021, and for older people and adults in April 2021. As a result, the fourth state of declared emergency ended on 30 September.

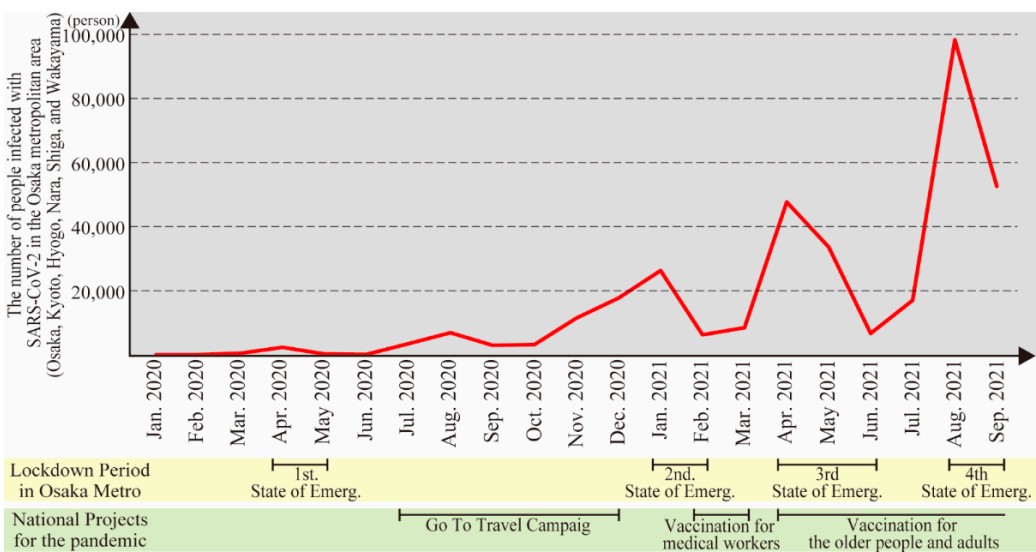

**Figure 2.** The number of people infected with SARS-CoV-2 in the Osaka metropolitan area.

### 2.3. Statical Analysis

This study analyzes the changes in the average population of the Osaka metropolitan area, ordinance-designed/regional hub cities, ordinary cities, and towns/villages from January 2017 to September 2021 over four different seasons. The population of the Osaka metropolitan area is the sum of the populations of ordinance-designed/regional hub cities, ordinary cities, and towns/village. The Osaka metropolitan area population is analyzed to understand the trend of all types of municipalities. The four seasons are classified as follows: winter is January, February, and March; spring is April, May, and June; summer is July, August, and September; and autumn is October, November, and December. In Japan, April is the year's turning point, starting school and jobs/careers. Therefore, the four seasons, set based on April, are suitable for analysis of Japanese population change.

This study calculated the average value for each season in each city. Then, the significant differences between seasons were clarified using the Wilcoxon signed-rank sum test. Then, significant statistical changes were analyzed using $p = 0.001$ as the criterion, presented in Section 3.1. For the statistical analysis, this study used JMP PRO 16.0 software.

A map of the number of population changes in each municipality during the urban exodus was shown in Section 3.2. Then, this study analyzed the relationship between population change and distance from the center of the Osaka metropolitan area. The center of the Osaka metropolitan area is the location of Osaka Station, which is the center of the business district. The results of this study show the characteristics of the locations of the municipalities where the urban exodus occurred.

### 2.4. Time Series Forecasting

This study forecasted the population trends from October 2021 to September 2022 based on the population data of the time series from January 2017 to September 2021. This study analyzed the time series forecasting of the Osaka metropolitan area, ordinance-

designed/regional hub cities, ordinary cities, and towns/villages. The future population is sometimes analyzed by the cohort component method rather than time series forecasting. The cohort component method estimates the future population of each cohort using the ratio of women and children and the net movement [35]. The cohort component method is suitable for forecasting stationary changes. However, because the urban exodus is a non-stationary change, this study analyzed it using time series forecasting.

For its forecasting, this study used state-space smoothing models. State-space smoothing models calculate the error, trend, and seasonal component by additive or multiplicative methods [36]. The trend component calculates changes in the long term, and the seasonal component calculates changes in the short term with repetition. The additive method is effective for linear changes, and the multiplicative method is effective for proportional changes. The trend component was calculated by not only the additive or multiplicative method but also the damped additive or the damped multiplicative method. For each time series, this analysis selects the best fitting model for forecasting based on the AIC (Akaike's information criterion) and BIC (Bayesian information criterion) scores. Smaller AIC and BIC scores indicate a better model fit. This study presents the best-fitting model's forecast value and interval in Section 3.3.

## 3. Results

### 3.1. Population Flow of Each Season

This section clarifies the changes in the population for each season in the Osaka metropolitan area of the ordinance-designed/regional hub cities, ordinary cities, and towns/villages from January 2017 to September 2021 in Figure 3. First, it was found that municipalities in the Osaka metropolitan area had a significant decrease in population before the COVID-19 pandemic in spring 2020. This result is the same as OECD Stat data [13], which show a population decline in the Osaka metropolitan area. However, no significant population change was found during the period from summer to autumn 2020. On the contrary, during that period, the population increased. In other words, there was a population inflow into the Osaka metropolitan area from outside the metropolitan area. However, after that period, the population began to decline significantly again. Therefore, population growth was a temporary phenomenon.

The result was the same for municipalities in ordinary cities and towns/villages. In other words, the average population of the municipalities in the ordinary cities and towns/villages decreased significantly before the COVID-19 pandemic in spring 2020. Then, during summer to autumn 2020, there was no significant population change, and the population increased. After autumn 2020, the population began to decrease significantly again.

However, the results were different for municipalities in ordinance-designated/regional hub and ordinary cities. Specifically, the average population of municipalities in ordinance-designed/regional hub cities decreased significantly from autumn to winter before the COVID-19 pandemic, on the back of having gradually decreased from 2017. However, the average population of municipalities in ordinance-designed/regional hub and ordinary cities also increased during the summer and autumn of 2020, when the population increased in ordinary cities and towns/villages. In particular, the population in the autumn of 2020 was similar to that in the autumn of 2017, the highest in the period analyzed. Subsequently, the population declined significantly during the winter and spring of 2021. Before the COVID-19 pandemic, the population declined during the autumn and winter months, indicating that the period of population decline was different between the pre-pandemic and the pandemic.

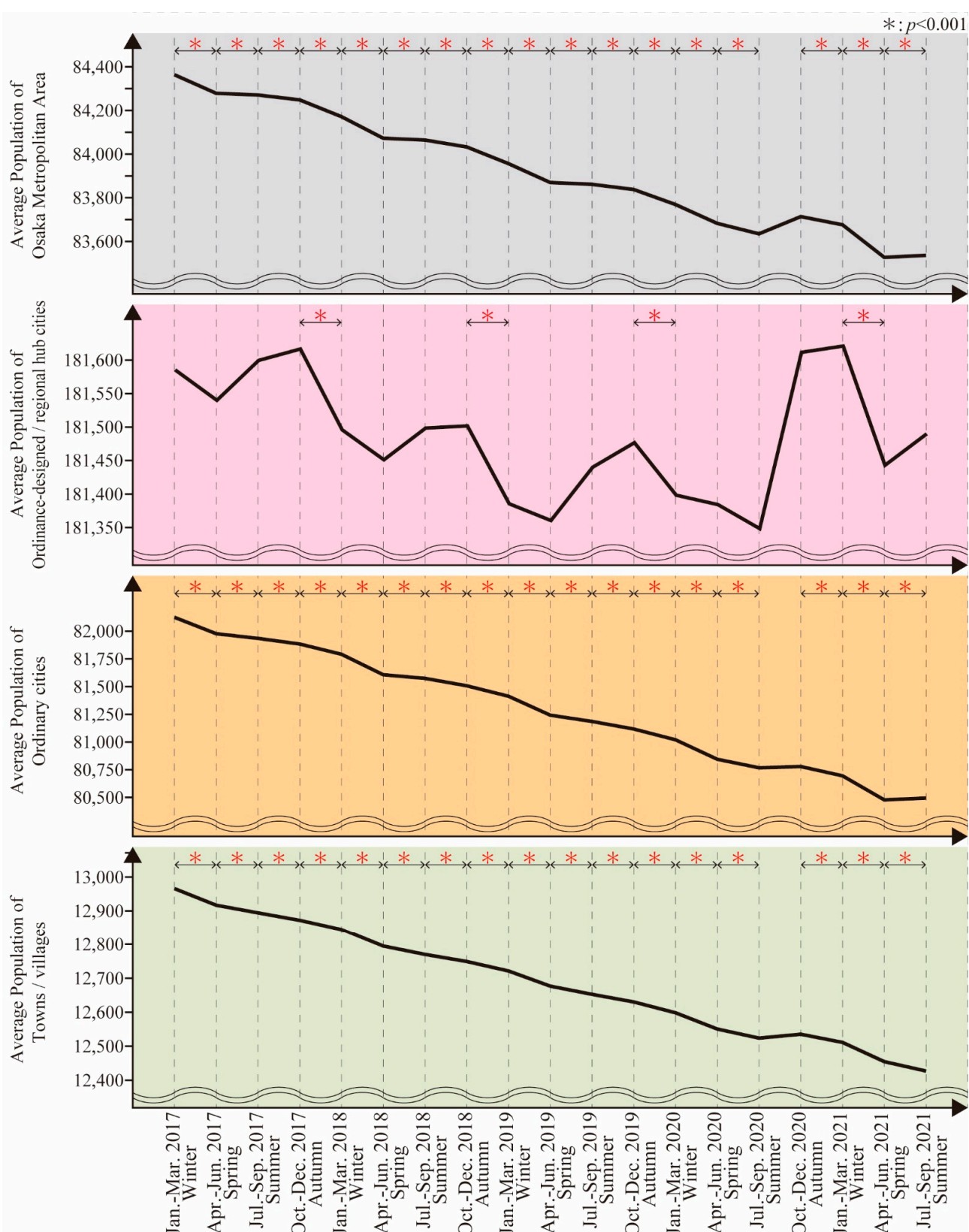

**Figure 3.** Population flow of each season.

### 3.2. Population Change during the Urban Exodus

　　Figure 4 shows the map of the average population changes in each municipality during the summer and autumn of 2020 when the urban exodus occurred. This study analyzed the

relationship between average population change (APC) of municipalities during the urban exodus and distance from the center of the Osaka metropolitan area. Figure 4 shows the spline curve and the confidence interval. The smoothing parameter of the spline curve λ was set to 0.05. The following paragraphs describe the results of the time-series prediction for the ordinance-designed/regional hub cities, ordinary cities, and towns/villages.

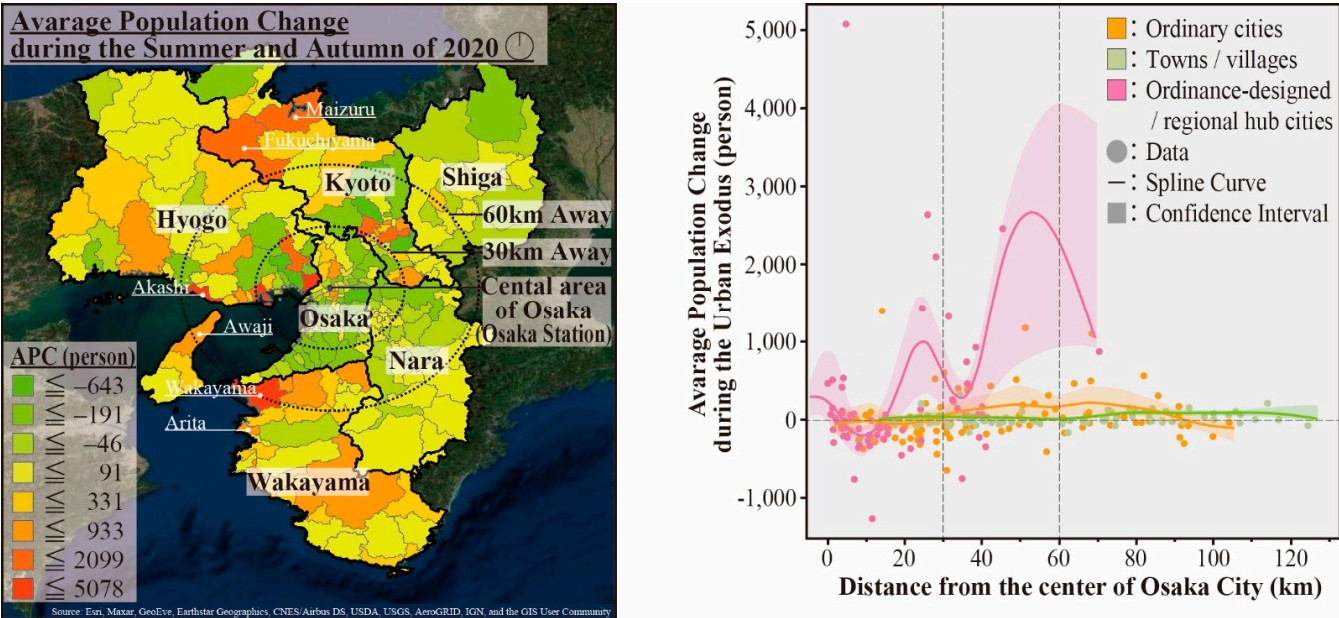

**Figure 4.** Population changes during the urban exodus.

It was found that the average population of ordinance-designed/regional hub cities decreased in the area of approximately 30 km. On the other hand, the average population increased over 30 km away from the city center. Examples are Wakayama City in Wakayama Prefecture (APC = 2460, 45.4 km) and Akashi City in Hyogo Prefecture (APC = 2813, 46.8 km). As such, the population increased in the ordinance-designed/regional hub cities.

It was found that the average population of ordinary cities decreased in the area of approximately 30 km. On the other hand, the population increased over 30 km away from the city center. Examples include Maizuru City (APC = 1108, 68.5 km) and Fukuchiyama City (APC = 1183, 51.2 km) in Kyoto Prefecture, Awaji City (APC = 518, 48.9 km) in Hyogo Prefecture, and Arita City (APC = 499, 68.2 km) in Wakayama Prefecture. As such, the population increased in the ordinary cities.

For the towns/villages, no relationship was found between the distance from the city center and population change.

### 3.3. Population Trend Forecasting

Figure 5 shows the total population trend from October 2021 to September 2022 based on the population data of the time series from January 2017 to September 2021 using state-space smoothing models. Based on the AIC and BIC scores, this study clarifies the better fitting model in the Osaka metropolitan area, ordinance-designed/regional hub cities, ordinary cities, and towns/villages in Table 1. The following paragraphs describe the results of the time-series prediction for the Osaka metropolitan area, ordinance-designed/regional hub cities, ordinary cities, and towns/villages.

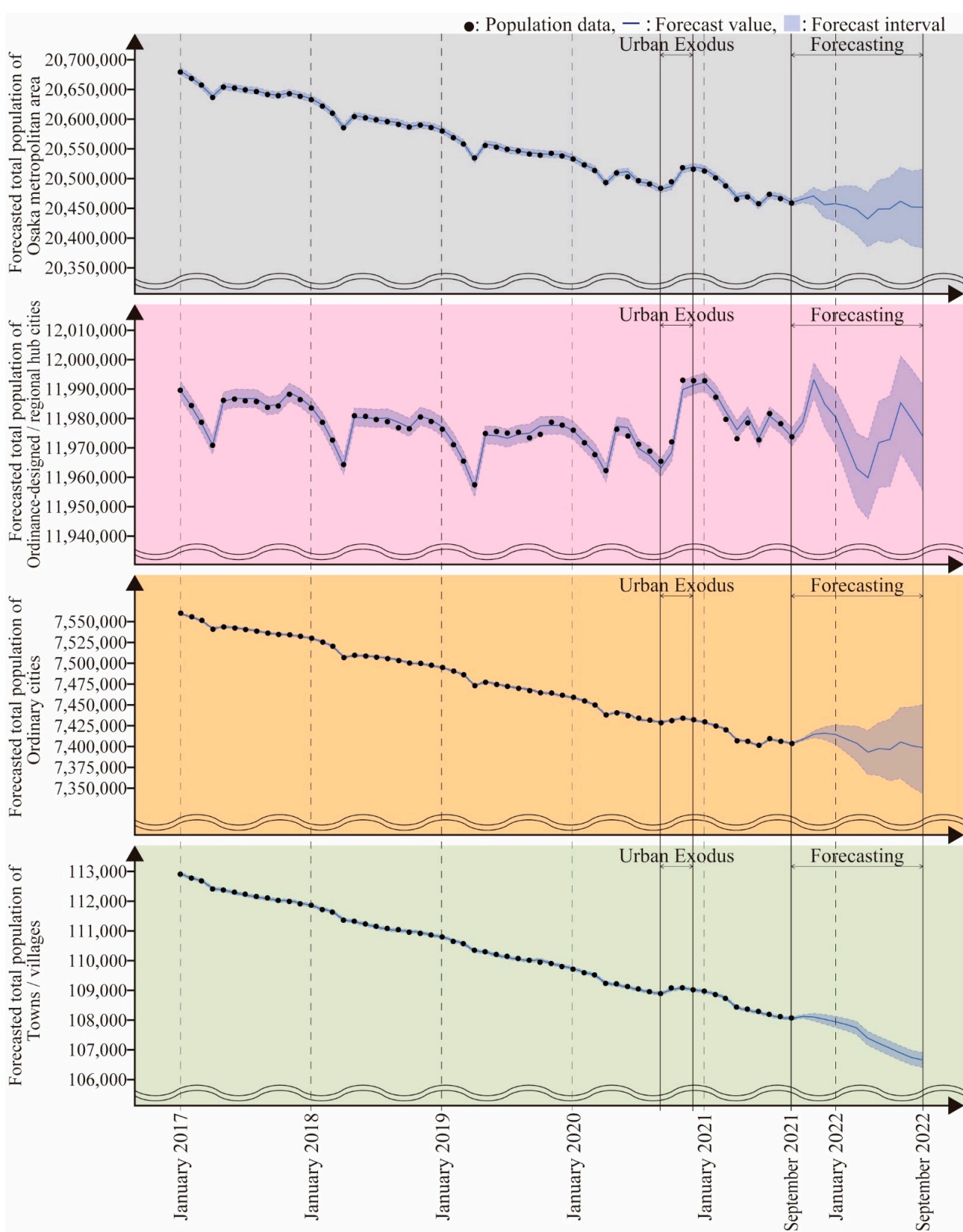

**Figure 5.** Population trend forecasting of each season.

**Table 1.** Statistics of the state–space smoothing models.

|  | Error | Trend Comp. | Seasonal Comp. | AIC | BIC |
|---|---|---|---|---|---|
| Osaka metropolitan area | M | Ad | M | 1159.174 | 1195.949 |
| Ordinance-designed/regional hub cities | A | Ad | M | 1083.841 | 1120.616 |
| Ordinary cities | M | Md | M | 1005.754 | 1042.529 |
| Towns/villages | M | A | M | 893.0206 | 927.7524 |

Note: A is the additive method; M is the multiplicative method; Ad is the damped additive method; Md is the damped multiplicative method.

In the Osaka metropolitan area, the state–space smoothing model was used as the multiplicative model for error, the additive damped model for the trend, and the multiplicative model for the seasonal component. Looking to the future, the forecast value shows that the speed of population decline will decrease. The forecast interval also suggests that the population might increase, although it might decrease as in the past.

In the ordinance-designed/regional hub cities, the state–space smoothing model was used as the additive model for error, the additive damped model for the trend, and the multiplicative model for the seasonal component. There was a significant population change in the ordinance-designed/regional hub cities even before the COVID-19 pandemic. Then, the ordinance-designed/regional hub cities also had significant population changes due to the urban exodus. Looking to the future, both the forecast value and interval suggest that the population change will be significant.

In ordinary cities, the state–space smoothing model was used as the multiplicative model for error, the multiplicative damped model for the trend, and the multiplicative model for the seasonal component. The forecast value and interval were found to be the same as for the Osaka metropolitan area. The forecast value shows that the speed of population decline will decrease. The forecast interval also suggests that the population might increase, although it might decrease as in the past.

In the towns/villages, the state–space smoothing model was used as the multiplicative model for error, the additive model for the trend, and the multiplicative model for the seasonal component. The forecast value and the forecast interval show that the population will continue to decline at the same rate as before the COVID-19 pandemic. In other words, in towns/villages, the population change has been unchanged by the pandemic.

## 4. Discussion

The results of this analysis indicate the impact of the urban exodus from the summer to autumn of 2020 in the Osaka metropolitan area. Due to the urban exodus, in ordinance-designed/regional hub cities, the population decreased in the area of approximately 30 km, and the population increased over 30 km away from the city center. The conclusion is important because the population increased not only in the ordinance-designed cities but also in the ordinance-designed/regional hub cities, unlike in the case of New York [10] and other metropolitan areas. That means that the urban exodus had the impact of increasing the population of the Osaka metropolitan area, which had been a continuously declining population. For both the ordinance-designed/regional hub and the ordinary cities, the most significant population increases occurred in the municipalities in the Osaka metropolitan fringe area, which are located more than 30 km away from the center. That means there was an inflow of population from outside the Osaka metropolitan area to municipalities on the Osaka metropolitan fringe area. In the fringe area, an increase in population provides the expected effects of maintaining the urban services necessary for people's daily lives, such as schools, medical and welfare facilities. The result is the new insights unique to the Osaka metropolitan area that this study clarified. For example, this occurred in the municipalities of Wakayama City in Wakayama Prefecture, Akashi City and Awaji City in Hyogo Prefecture, and Maizuru City and Fukuchiyama City in Kyoto Prefecture. In the future, the

population trends might exacerbate the shrinking of ordinance-designed/regional hub and ordinary cities. Those findings are summarized in Table 2.

**Table 2.** Summary of population changes in each type of city.

|  | Ordinance-Designed /Regional Hub Cities | Ordinary Cities | Towns /Villages |
|---|---|---|---|
| Population changes before the COVID-19 pandemic | Decline | Decline | Decline |
| Population changes during the urban exodus (Summer to Autumn 2020) | Increase | Increase | Increase |
| Population trend forecasting (October 2021 to September 2022) | Increase or Decline | Increase or Decline | Decline |
| Population growth cities during the urban exodus | Wakayama Akashi | Awaji Maizuru Fukuchiyama | |

In the summer and autumn of 2020, there was no outbreak of the new variant of SARS-CoV-2 in Japan, and the government tried to balance economic recovery with the prevention of the infection's spread. With the declaration of a state of emergency from April, work-from-home took hold, and careers changed. For example, a change in human mobility was observed between April 2019 and April 2020 in Ibaraki City, an ordinary city in the Osaka metropolitan area [37–39]. The work-from-home trend might have made it easier to choose to migrate in the medium term. Therefore, during the pandemic, people tend to choose where to live based not only on their workplace but also on the surrounding green environment and medical facilities [16,17]. Even before the pandemic, this trend was noticed in the residence choice preferences of specific residents, such as older adults [40] and migrants to their hometowns [41]. However, the pandemic expanded the population who changed residential location choice preferences.

Regarding the origin area of the urban exodus, this study clarified the inflow of population from outside the Osaka metropolitan area. The Japanese Cabinet Office [42] conducted a web questionnaire survey in April–May 2021 and reported that more young people considered emigrating from the central Tokyo metropolitan area. The Hometown Return Support Center [43], a non-profit organization that supports migration in Tokyo, reported that a seminar on migration organized by the Wakayama Prefecture had the highest number of participants in 2020. Based on those surveys, the population inflow to the Osaka metropolitan area might be understood to have come from the Tokyo metropolitan area. In the Tokyo metropolitan area, excessive-high population density has been considered a problem. For example, some companies have relocated some of their headquarters' functions from the Tokyo metropolitan area to the Osaka metropolitan area [44]. A decrease in the population of urban centers would have the effect of preventing an increase in the number of infected people with the SARS-CoV-2. Furthermore, it would have the effect of preventing localized high densities and maintaining a well-balanced density of locations in Japan as a whole.

## 5. Conclusions

In conclusion, this study clarified the possibility of population growth in shrinking ordinance-designed/regional hub and ordinary cities by the urban exodus triggered by the COVID-19 pandemic. The results suggest that it might be possible to prevent population decline in many cities in the Osaka metropolitan area, where population decline has been predicted to occur. The urban exodus contributes to the need for the local governments of shrinking cities to maintain the urban services necessary for people's daily lives, such as schools and medical and welfare facilities. It indicates the importance of considering the impact of urban exodus on the future population of shrinking cities, in addition to the economic decline, political change, and aging society that have been previously discussed.

However, the population increase in the Osaka metropolitan area may not be the same in all metropolitan areas. This is because the population may have decreased in the Tokyo metropolitan area. Therefore, it is necessary to study where the population came from. Japanese census data does not include where the immigrants previously lived to protect personal information. Therefore, questionnaire survey can be conducted for those who have moved to the Osaka metropolitan fringe areas during the urban exodus because.

The limitation of this study is that we do not know the driving factor of the urban exodus, which is the change of people's residential location choice preferences. Some previous studies found that people's residential location preferences changed from workplaces to medical and welfare facilities and the surrounding natural environment [16,17]. Besides, some local governments, where increased the population during the urban exodus, have enhanced their policy programs, such as attracting companies, providing part of the cost of migration, and supporting childcare [45]. The Osaka metropolitan area can prevent a rapid population decline in the long term if future studies will clarify the change in people's residential location choice preferences and examine methods of introducing policies even in cities with little financial resources. For instance, policies that improve walkability are effective not only in maintaining the population [46] but also in improving urban sustainability [47]. Future research should clarify effective measures for more cities to shift to population growth in the Osaka metropolitan area.

**Author Contributions:** Conceptualization, H.K.; methodology, H.K.; software, H.K.; validation, H.K. and A.T.; formal analysis, H.K.; investigation, H.K.; resources, H.K.; data curation, H.K.; writing—original draft preparation, H.K.; writing—review and editing, H.K. and A.T.; visualization, H.K.; supervision, H.K.; project administration, H.K.; funding acquisition, H.K. and A.T. All authors have read and agreed to the published version of the manuscript.

**Funding:** This research was funded by the JSPS KAKENHI (grant number 21K14318).

**Institutional Review Board Statement:** Not applicable.

**Informed Consent Statement:** Not applicable.

**Data Availability Statement:** The data presented in this study are available from references [25–30].

**Conflicts of Interest:** The author declares no conflict of interest.

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
