# Peer review of "Impact of the Urban Exodus Triggered by the COVID-19 Pandemic on the Shrinking Cities of the Osaka Metropolitan Area"

_sustainability, doi:10.3390/su14031601_

Round 1
Reviewer 1 Report
The topic of the urban exodus triggered by the COVID-19 Pandemic in this study is interesting. A study of the epidemic impact on the shrinking cities of the Osaka metropolitan area and possible trends should be of reference for policy making.
However, in the Discussion and Conclusion part, it is recommended that the authors strengthen the analysis of the internal reasons of how the COVID19 epidemic impacts population changes, especially the population increasing in the fringe area, and provide more specific and credible explanations, and on this basis, make more relevant policy recommendations.
Author Response
Dear Reviewer:
We appreciate the reviewer for the generous comment on the manuscript. We have attached our response letter in PDF format. We believe that the manuscript is now suitable for publication in Sustainability and look forward to hearing from you concerning your decision.
Yours sincerely
Haruka Kato

Reviewer 2 Report
Dear Authors,
I hope you're well
I appreciate your scientific effort to produce this manuscript and I have some comments to help you improve it:
1- The abstract should state the major points of your research briefly and explain why your work is important your purpose, how you went about your project, what you learned, and what you concluded. An abstract is often presented separately from the article, so it must be able to stand alone.
2- No need for sub-sections in the introduction as it serves the purpose of leading the reader from a general subject area to a particular field of research. It establishes the context of the research being conducted by summarizing current understanding and background information about the topic, stating the purpose of the work in the form of the hypothesis, question, or research problem, briefly explaining your rationale, methodological approach, highlighting the potential outcomes your study can reveal, and describing the remaining structure of the paper.
3- The discussion part should be merged into the results section " Results and Discussion". The purpose of the discussion section is to interpret and describe the significance of your findings in relation to what was already known about the research problem being investigated and to explain any new understanding or insights that emerged as a result of your research.
4- The "Conclusions" section intends to help the reader understand why your research should matter to them after they have finished reading the paper. It is suggested to organize the "Conclusion" section much better. This section should be presented in one 250-300 words paragraph that contains unique results and findings. Also, make sure your conclusions' section underscores the scientific value-added of your paper.
5- The paper language and punctuation need revision by an expert.
6- It is not recommended to use a pronoun (i.e. we and our) when writing research papers. Formal writing is almost always written in the third person.
Author Response

(The authors gave the same response as above.)

Reviewer 3 Report
I like the manuscript.
Yet, some issues need to be improved.
- Check the writing across the whole paper. Some sentences should be corrected.
Example:
Lines 35 and 36…. The COVID-19 pandemic and ……. the COVID-19 epidemic: there is a lack of consistency. There is also unnecessary repetition.
Line 60…… is helpful example to promote other countries to consider relaxing their restrictions. Here the authors are using inappropriate words: the idea can be grasped, but it is not well expressed.
- Methodology: elaborate on how the literature review was carried out
- The literature review should be improved so that the issues of urban exodus could be largely discussed, (including the driving factors, the impacts/effects or consequences in both areas of the origin and destination).
- Write down the contribution of your research to the scientific knowledge or its policy implications in both Japan and other developed countries facing similar trend of shrinking cities.
Author Response

(The authors gave the same response as above.)

Reviewer 4 Report
The article addresses a very important research problem of the shrinkage of cities during the COVID-19 pandemic. It is thus a relevant and new take on a fairly well-studied research problem. The methodology chapter is well-outlined. The research results are concisely presented. However, attention should be brought to the disorganized structure of the article: the "Discussion" and "Conclusion" chapters are combined, and it is suggested that they should be split and re-edited, as currently the joint chapter is largely a discussion chapter. It is advisable to move and further explore some of the discussed issues to the newly created "Conclusion" chapter (it should include, among others, conclusions on the prospects for further research). There is no doubt that a typical summary chapter with clearly defined conclusions is missing in the paper. In addition, the subchapters in "Introduction" should be removed, especially given it is a fairly short chapter. References could also be expanded to include new publications.
Technical remarks: Figures 1 and 4 mention no source of the basemap used for creating the maps. Moreover, it is suggested that national borders should be marked in Figure 1.
Author Response

(The authors gave the same response as above.)

Round 2
Reviewer 4 Report
Figure 1 should be corrected in terms of:
- a clear demarcation between North Korea and South Korea,
- entering the name of the capital of North Korea in English.
Author Response

(The authors gave the same response as above.)
